



*

# The development of rainfall retrievals from radar at Darwin

Robert Jackson[1], Scott Collis[1], Valentin Louf[2], Alain Protat[3], Die Wang[4], Scott Giangrande[4], Elizabeth J. Thompson[5], Brenda Dolan[6], and Scott W. Powell[7]

[1]Argonne National Laboratory, 9700 Cass Ave., Lemont, IL, USA
[2]School of Earth, Atmosphere and Environment, Monash University, Clayton, VIC, Australia
[3]Bureau of Meteorology, 700 Clayton St., Docklands, VIC, Australia
[4]Brookhaven National Laboratory, 98 Rochester St., Upton, NY, USA
[5]National Oceanic and Atmospheric Administration Physical Sciences Laboratory, 325 Broadway, Boulder, CO, 80305
[6]Colorado State University, Department of Atmospheric Sciences, 3915 W Laport Ave, Fort Collins, CO 80523
[7]Department of Meteorology, Naval Postgraduate School, Monterey, CA

**Correspondence:** Robert Jackson (rjackson@anl.gov)

**Abstract.**

The U.S. Department of Energy Atmospheric Radiation Measurement program Tropical Western Pacific site hosted a C-band POLarization (CPOL) radar in Darwin, Australia. It provides two decades of tropical rainfall characteristics useful for

validating global circulation models. Rainfall retrievals from radar assume characteristics about the droplet size distribution (DSD) that vary significantly. To minimize the uncertainty associated with DSD variability, new radar rainfall techniques use dual polarization and specific attenuation estimates. This study challenges the applicability of several specific attenuation and dual-polarization based rainfall estimators in tropical settings using a 4-year archive of Darwin disdrometer datasets in conjunction with CPOL observations. This assessment is based on three metrics: statistical uncertainty estimates, principal

component analysis (PCA), and comparisons of various retrievals from CPOL data.

The PCA shows that over 99% of the variability in estimated rainfall rate $R$ can be explained by radar reflectivity factor for rainfall rates $1 < R < 10\ mm\ hr^{-1}$. These rates primarily originate from stratiform clouds and weak convection (median drop diameters less than 1.5 mm). The dual-polarization specific differential phase increases in usefulness for rainfall estimators in times with $R > 10\ mm\ hr^{-1}$. Rainfall estimates during these conditions primarily originate from deep convective clouds with

median drop diameters greater than 1.5 mm. Using specific attenuation for estimating $R$ generally does not provide additional skill beyond other metrics for Darwin. An uncertainty analysis and intercomparison with CPOL show that a CSU-blended technique for tropical oceans, with modified estimators developed from VDIS observations, is most appropriate for use in all cases, such as when $1 < R < 10\ mm\ hr^{-1}$ (stratiform rain), and when $R > 10\ mm\ hr^{-1}$ (deeper convective rain).



## 1 Introduction

Accurate rainfall accumulation and rate estimates are crucial for validating global circulation model (GCM) simulations of precipitation. A known problem of many GCMs, including the U.S. Department of Energy's Earth Energy Exascale System Model (E3SM) (Golaz et al., 2019), is that the phase and magnitude the diurnal cycle of precipitation are not adequately

resolved due to the parameterization of convection (Del Genio, 2012). Multidecadal datasets, such as those recorded at the U.S. Department of Energy's Atmospheric Radiation Measurement (ARM) program's Tropical Western Pacific (TWP) site in Darwin, Australia (Keenan et al., 1998; Mather et al., 2016; Long et al., 2016), provide unique opportunities to develop climatologies and process-level parameterization constraints for GCM simulations. For example, Kumar et al. (2013), Rauniyar and Walsh (2016) and Jackson et al. (2018) have previously developed climatologies of radar-estimated cloud top heights from

17 years C-band POLarization (CPOL) data at the ARM TWP site to be used for validation of E3SM. Dolan et al. (2013) also identified the hydrometeor types present in the clouds and precipitation sampled by CPOL over seven seasons.

    As with the previous cloud top height and hydrometeor type constraints, accurate rainfall accumulations and rates are vital quantities to evaluate or improve in models such as in convective parameterizations in E3SM (Tang et al., 2019) and other general circulation models for addressing uncertainties in their predictions. However, developing a suitable rainfall rate cli-

matology or similar summary rain accumulation statistics from a single radar dataset is a nontrivial task. Several different methodologies exist for measuring or estimating rainfall accumulations and/or instantaneous rainfall rates. For instance, at the ARM TWP site, rain gauges and disdrometers provide estimates of rainfall rate and collect individual particle statistics. Even if a perfect rainfall estimation method is determined, using such data for climatological scale analysis is complicated by the fact that the rainfall experienced at TWP is not representative of the spatial variability in rainfall rate over GCM-scale domains

$O([20-100km])$. In contrast, scanning radars such as the CPOL may estimate rainfall accumulations over this wider spatial domain $O([100-150km])$ with horizontal resolution of $O(100m)$ (e.g., Keenan et al. (1998)). However, scanning radars typically retrieve $R$ using empirical power law relationships between $R$ and the radar observables, which may be subject to uncertainty contingent on the representativeness of these fits. These relations often utilize conventional radar quantities such as the radar reflectivity factor $Z_h$, and dual-polarization radar quantities such as specific differential phase $K_{dp}$, and differ-

ential reflectivity $Z_{dr}$. Single-moment and blended empirical relationships are commonplace in the literature (i.e. Marshall and Palmer (1948); Aydin and Giridhar (1992); Ryzhkov and Zrnić (1995); Matrosov (2005); Matrosov et al. (2006); Wang et al. (2013); Ryzhkov et al. (2014); Thompson et al. (2015, 2018); Wang et al. (2018); Giangrande et al. (2019)) and typically developed from simulated radar moments informed by the drop size distributions (DSDs) sampled by disdrometers at various global locations. Recently, studies have attempted to combine the advantageous dual-polarization radar measurement

properties more seamlessly into a single radar quantity by estimating the specific attenuation $A_h$ for similar empirical rainfall applications (e.g., Ryzhkov et al. (2014); Giangrande et al. (2014b)). Dual-polarization relationships have traditionally been the preferable option for radar rainfall rate estimates, as these have been found less sensitive to potential biases owing to DSD variability, radar miscalibration, partial beam blockage, and/or attenuation in rain (Doviak and Zrnić, 1993; Bringi and Chandrasekar, 2001; Ryzhkov et al., 2005). Nevertheless, single and/or dual-polarization power law relationships are often sensitive



to the underlying differences found in the limits of the DSD observations used to develop those relationships, which vary over different regions of the globe due to the changing nature for dominant cloud dynamical/microphysical processes (Bringi et al., 2003, 2009; Dolan et al., 2018). Similar issues are also anticipated for newer radar-rainfall algorithm concepts such as machine learning efforts using Neural Network or Gaussian mixtures concepts (e.g. Vulpiani et al., 2009; Li et al., 2012) that are trained

and/or validated within comparable DSD observational limitations.

An important consideration for applying radar rainfall methods to different regions across the globe is that the majority of aforementioned rainfall studies have emphasized the properties of midlatitude continental clouds, and often over relatively modest data records (i.e. 10s of events). For many practical hydrological applications, the best references are those for NOAA's S-band (10-cm wavelength) NEXt Generation RAdar (NEXRAD), with most operational relations weighted towards Okla-

homa, Florida, Colorado and/or deeper convective cloud conditions (Ryzhkov et al., 2005). The relative absence of extended, ground-based rainfall retrieval validation datasets outside of midlatitude regions poses several challenges for potential global rainfall applications and possible model evaluation. DOE ARM operates multiple fixed sites in distinct global regimes, including the Darwin TWP, the Oklahoma Southern Great Plains (SGP) (Sisterson et al., 2016), and Eastern North Atlantic (ENA) site in the Azores (Giangrande et al., 2019). Prior studies that focused on Darwin (Keenan et al., 1998; Bringi et al., 2003, 2009;

Thurai et al., 2010) indicate that midlatitude $R$ esimators and DSD variability are less applicable outside of the midlatitudes.

This study focuses on $R$ estimators at C-band and X-band for climatological studies and model improvement over Darwin, Australia. Specifically, the ARM TWP site hosts C/X-band (5 cm/3 cm wavelength) Scanning ARM Precipitation Radars (C/XSAPRs) and the C-band Polarization (CPOL) radar (Keenan et al., 1998). Recently, Giangrande et al. (2014b) found that for the ARM SGP site CSAPR during the MC3E campaign, $K_{dp}$-based retrievals generally provide an optimal estimate of

rainfall for accumulations greater than 10 mm when compared to $A_h$-based retrievals. For the tropical oceans, Thompson et al. (2018) showed that the root mean square error between the disdrometer and radar estimated $R$ at C-band and X-band was lowest when the Colorado State University (CSU) blended technique for tropical oceans was used with relations formed from tropical DSD measurements. The algorithm, originally developed by Cifelli et al. (2011), uses $Z_h$, $Z_{dr}$ and/or $K_{dp}$ as input depending on whether values of these fields are significantly above noise thresholds. In the same study, RMSE was slightly

higher for $R$ estimated by $A_h$ at these wavelengths. However, these retrievals were developed and using data over Manus and Gan Islands. These small atolls experience open-ocean conditions, such that large raindrops from melted hail were rare, even in strong convection (Thompson et al., 2015). For the ARM TWP site in Darwin, deep mixed-phase convection, formed by seabreeze convergence, is common (Rutledge et al., 1992; Williams et al., 1992; May and Rajopadhyaya, 1999; Kumar et al., 2013; May and Ballinger, 2007; Jackson et al., 2018). Therefore, there is greater potential for the impact of cold rain processes

(melted hail or graupel that forms large droplets) on determining the surface DSD in Darwin compared to Manus and Gan Islands.

It remains unclear whether, or which, prior $R$ estimators are completely suited to the challenges of creating accurate multi-decadal, climate-research quality datasets at TWP Darwin at C- and X-band. Lately, more radar rainfall estimators at shorter wavelengths have been developed. However, these efforts typically have access to data from relatively short field campaigns

or a handful of case studies of extreme events. In this regard, these efforts are valuable but potentially not well-matched to the





challenges of creating multidecadal datasets at TWP Darwin. This and the lack of studies at sites with a mixture of typical and extreme rainfall events stresses the importance of further assessing $R$ retrievals for CPOL and other ARM radars at the ARM TWP site. To accomplish this task, this study uses four years of co-located two dimensional video disdrometer (VDIS) and CPOL data at the ARM TWP site, providing a longer and therefore hopefully more representative dataset than used in prior

Darwin-based studies.

This study is organized as follows. To assess the applicability of dual polarization quantities and an additional estimated quantity $A_h$ for retrieving rainfall rate $R$ for ARM radars in Darwin, this study will use simulated radar observables generated from the VDIS observations collected during tropical convective events over the ARM TWP site. Data and methods are introduced in Section 2. Observational results from these data are shown in Section 3. Section 4 assesses the importance of $Z_h$, $Z_{dr}$,

$K_{dp}$, and $A_h$, in determining or informing the rainfall rate estimates over this dataset using principal component analysis (PCA) on these quantities. These analyses are performed at the C- and X- band radar wavelengths utilized by the ARM program. To evaluate C-band retrievals, Section 4 also compares VDIS rainfall rates against retrievals from CPOL data at the ARM Darwin TWP site. Section 5 includes the main conclusions of this study.

## 2   Datasets

### 2.1   CPOL

The C-band polarization radar (CPOL) (Keenan et al., 1998) provided plan position indicator (PPI) scans of $Z$, $Z_{dr}$, and differential phase $\phi_{dp}$ at elevations of 0.5°, 0.9°, 1.3°, 1.8°, 2.4°, 3.1°, 4.2°, 5.6°, 7.4°, 10.0°, 13.3°, 17.9°, 23.9°, 32.0°, and 43.1° every 10 minutes from 1998 until 2017 except during 2008 and 2009. Data from the 2011 to 2015 seasons are used in this study, which correspond to the times VDIS observations were available at the ARM TWP site in Darwin. In total, these

datasets correspond to a window in time over which the Darwin location recorded 4884 mm of rainfall. CPOL provides radar variables at a 250 m along-gate resolution and a 1° azimuthal resolution. The maximum unambiguous range of CPOL is 150 km. The Python ARM Radar Toolkit (Py-ART) was used to process and visualize the CPOL data (Helmus and Collis, 2016). Clutter and second trip echoes were removed using a technique based on the texture of the Doppler velocity field previously applied to CPOL data by Jackson et al. (2018).

For rainfall retrievals, a robust calibration and attenuation correction of $Z_h$ and $Z_{dr}$ are paramount. Therefore, in this study, $Z_h$ was calibrated using the Relative Calibration Adjustment technique (Wolff et al., 2015), previously applied to CPOL data, that integrates the use of ground clutter with space borne radar observations and the self consistency of the polarimetric radar moments to monitor for changes in the radar calibration (as for Darwin CPOL, see Louf et al. (2019)). In addition, $Z_h$ and $Z_{dr}$ at C-band are prone to (differential) attenuation from heavy rainfall which may bias (underestimate) $R$. Therefore, we apply the

Z-PHI method in order to adjust $Z_h$ and $Z_{dr}$ for attenuation due to heavy rain. This Z-PHI approach also provides an estimate of the specific attenuation $A_h$ (Gu et al., 2011). The $\phi_{dp}$ was dealiased in order to ensure that it monotonically increases with range. We then applied a linear programming phase processing technique of Giangrande et al. (2013) to estimate the $K_{dp}$ from these dealiased $\phi_{dp}$ profiles.



## 2.2 Disdrometers

Disdrometers are the primary method by which rainfall rates and DSDs parameters are recorded in this study. Previous disdrometer efforts at the ARM TWP Darwin site have explored the extended Darwin Joss-Waldovel (J-W) disdrometer record (e.g. Giangrande et al., 2014a). However, J-W disdrometers are potentially less optimal for assessing dual-polarization radar efforts in lighter rain and/or small-drop conditions. Recently, ARM ENA disdrometer comparisons suggested that the 2D Video Disdrometer (VDIS, also referred to as 2DVD) provided improved estimates of the DSD in light rain (Giangrande et al., 2019) compared to the J-W disdrometer. Therefore, we opt to explore the VDIS record to characterize the DSDs and perform subsequent dual-polarization radar quantity calculations from them. In order to ensure quality DSDs, artifacts that are due to splashing or other causes need to be removed. Following Giangrande et al. (2019) and Wang et al. (2018)'s analysis of VDIS DSDs, thresholds that check drop fall speed and particle diameter were applied to filter out splashing. After that, the DSDs were averaged to 1 minute to reduce noise and then fitted to a normalized gamma distribution of the form $N(D) = N_w F(D/D_0)$ determined by two parameters $N_w$ and $D_0$ (Testud et al., 2001). These fits were produced using the method of moments technique in PyDSD (Ulbrich and Atlas, 1998; Hardin and Guy, 2017) utilized in past ARM efforts analyzing VDIS and J-W DSDs (Giangrande et al., 2014a, 2019; Wang et al., 2018). In order to ensure a statistically significant sample required to calculate the gamma distribution parameters, only DSDs with greater than 100 drops and rainfall rates greater than $0.5 \ mm \ hr^{-1}$ were included. After these thresholds, 35 211 raining minutes of rain rate and DSD data remained available for use in this study. Changing the drop number threshold to 50, 200, and 500 did not significantly impact the results that follow.

## 2.3 Radar moment simulations from DSD

For each of the 1-minute DSDs in the VDIS dataset shown in Figure 1, the simulated observables $Z_h$, $Z_{dr}$, $K_{dp}$ and $A_h$ were calculated by performing T-matrix scattering simulations (Mishchenko et al., 1996) at C and X-band using Py-TMatrix and PyDSD (Leinonen, 2014; Hardin and Guy, 2017) that has been utilized in past ARM efforts Wang et al. (2018); Giangrande et al. (2019). A drop shape model is required for these simulations. We used Brandes et al. (2002)'s drop size model and a standard deviation of the canting angle of 12°, following Louf et al. (2019)'s CPOL $Z_h$ calibration. The air temperature was assumed to be 20°C for all of the simulations similar to tropical surface air conditions at Darwin. $A_h$ estimates appeared physically consistent with measurements of liquid water content $W$, which are related to either nearly or exactly the 3rd moment of the DSD, respectively.

## 3 DSD observations and simulated radar variables from DSD

### 3.1 DSD parameters

Figure 1 shows $N_w$, $D_0$, and $R$ estimated from the VDIS for all DSDs considered. The $D_0$ distribution is right-tailed, and spans values from 0.5 mm up to 4.5 mm. 90% of the $D_0$ values are less than 1.8 mm. The $N_w$ spans 5 orders of magnitude and $R$ reached up to $150 \ mm \ hr^{-1}$. Much of this DSD variability is typically attributable to differences in whether the precipitation





originates from stratiform or convective clouds and how great of a role is played by ice-based or mixed phase precipitation (i.e. Tokay and Short, 1996; Bringi et al., 2003, 2009; Thompson et al., 2015; Dolan et al., 2018). Therefore, it is important to stratify the DSD data by whether they were produced by stratiform and convective clouds. However, different studies have defined stratiform and convective clouds using different thresholds for classification depending on cloud conditions. Bringi

et al. (2003), Bringi et al. (2009), and Giangrande et al. (2014a), applied a two-moment DSD-based classification to Darwin datasets such that any DSD having $\log_{10} N_w > 6.3 - 1.6 D_0$ is labelled as convective, whereas remaining DSDs are marked as stratiform. This relationship (herein, BR09) was developed using disdrometer data in Darwin, Colorado, and other land locations, enabling deep convective rainfall to be distinguished from widespread stratiform precipitation.

More recently, Thompson et al. (2015) (T15) proposed a definition for Manus and Gan Island that classifies all DSDs with

$\log_{10} N_w > 3.8$ as convective, which was consistent with prior work by Bringi et al. (2003, 2009) and Thurai et al. (2010). The T15 definition separated the full range of tropical oceanic convection (weak to strong) from stratiform precipitation. Confirmation for these T15 DSD separations have been subsequently reported by Giangrande et al. (2019) using disdrometers coupled with cloud radar over the ARM oceanic-mid-latitude ENA facility and by Dolan et al. (2018) with other datasets around the world. Nevertheless, both Thompson et al. (2015)'s oceanic dataset and recent ENA findings do not cover continental-based,

deep ice-based convection with hail. Such mixed-phase deep convection and organized convective systems are common over the Darwin region (Williams et al., 1992; Rutledge et al., 1992; May and Rajopadhyaya, 1999; May and Ballinger, 2007; Bringi et al., 2009; Thurai et al., 2010; Jackson et al., 2018). Prior studies find that tropical-oceanic cloud behaviors do not solely drive most of the surface rainfall here, particularly when air flow is from land-to-sea or has significant land-influence. These considerations may be analogous to other displays for tropical-continental conditions (Tokay and Short, 1996; Wang

et al., 2018), wherein BR09 is typically sufficient to distinguish deeper convective cores from other forms of precipitation. Therefore, this study applies the BR09 convective-stratiform classification to distinguish between the strong convective DSDs and other rain types. We isolate and focus on deep convection in order to study the phenomena most likely to contribute to strong magnitude variability in $R$. As shown by Thompson et al. (2015) and Dolan et al. (2018), DSDs not classified as convection by BR09 could include contributions from both weak ocean-based convection in addition to stratiform clouds.

Here, we simply refer to all non-convective rain classified by BR09 as stratiform.

Figure 2 shows normalized frequency histograms of $N_w$, $D_0$, and $W$, separated by the B09 classification. In addition, summary statistics of these variables are given in Table 1. There are 35 211 DSDs in total that fit the filtering criteria used to generate Figure 1. The BR09 classification indicated that 26 131 of these DSDs were not convection. In total, 750 mm, or 21% of the total rainfall accumulation, originated from stratiform rain (Table 1). Past studies in Darwin by Tokay and Short (1996)

and Giangrande et al. (2014b) reported that about 30% of the total rainfall accumulation originated from stratiform clouds. Their data and the data here are consistent with the notion that rainfall in Darwin primarily originates from convection. For these stratiform DSDs, $W$ is generally less than $1\ g\ m^{-3}$ (Figure 2 and Table 1), from which less attenuation of the radar beam by liquid water is expected and quantified by T18. The $W$ and $D_0$ values in Table 1 are lower in stratiform DSDs compared to convective DSDs. The smaller drops in stratiform DSDs for given $R$ shows that these DSDs more likely originated from crystal



aggregation aloft in stratiform rain devoid of melting hail (Thurai et al., 2010; Dolan et al., 2018). These relative differences in $W$, $D_0$ and $N_0$ have been shown in Darwin previously (Thurai et al., 2010; Giangrande et al., 2014a).

Convective DSDs exhibited right-tailed distributions of $D_0$ and left tailed distributions of $N_0$ and $W$ (Figure 2). In Figure 1, the right tail of $D_0$ is associated with lower $N_0$ and $W$, consistent with fewer and larger drops. This tail has been observed in

previous studies in Darwin (Giangrande et al., 2014a) as well as in other regions such as the Amazon (Wang et al., 2018) and the ARM ENA site (Giangrande et al., 2019). This tail is likely caused by different stages of deep convection being sampled. Large hail grown by accretion that then melts and falls to the ground has very low $N_0$ and very high $D_0$ (Bringi et al., 2003, 2009). At the edges of deep convective clouds, size sorting favors fewer, but much larger drops hitting the ground before the more numerous smaller drops do (Gunn and Marshall, 1955; Thompson et al., 2015). In addition, for given ranges of $R$,

there are lower values of $N_w$ and higher values of $D_0$ in Table 1 for convective DSDs compared to stratiform DSDs. This is consistent with land-based convective-stratiform classification proposed by BR09 using data from Darwin and also other mid-latitude regions such as Colorado.

When looking at histograms of $R$ in Figure 2, it is clear that there is some overlap between convective and stratiform rain DSDs (the stratiform category of BR09 could also include weak oceanic convection, T15). We find that, 95.7% of the DSDs

with $R > 10\ mm\ hr^{-1}$ are classified as convective, while 9.7% of the DSDs with $R < 10\ mm\ hr^{-1}$ are convective. This is consistent with prior studies of tropical rain (T15, Rutledge et al., 2019). Therefore this Darwin data shows that the majority of cases with $R < 10\ mm\ hr^{-1}$ are likely produced by stratiform rain and weak convection while the cases $R > 10\ mm\ hr^{-1}$ are likely the result of deep convection. Therefore, the DSDs collected here are the result of different modes of rain drop formation. Warm rain processes in narrower DSDs are more likely present when $R < 10\ mm\ hr^{-1}$ (T15). Meanwhile, deeper convection

where cold rain processes (melting hail) occur is more likely to be present during times when $R > 10\ mm\ hr^{-1}$. Past studies have used $R = 10\ mm\ hr^{-1}$ as a threshold for separating convective and stratiform DSDs (Tokay and Short, 1996; Nzeukou et al., 2004), so this separation threshold is consistent with past literature examining DSDs in Darwin. However, as shown here and in Giangrande et al. (2014a), thresholds based on $R$ exclude some weak convective events and the lateral edges of strong convection when number concentration is still low.

## 3.2  Simulated radar moments from DSD

Figure 3 shows scatter plots of $R$ as a function of simulated $Z_h$, $K_{dp}$, and $A_h$ at C- and X-band from the VDIS DSDs in Figure 1. In addition, consistent with what has been done in many past studies (i.e. Marshall and Palmer (1948); Aydin and Giridhar (1992); Matrosov (2005); Matrosov et al. (2006); Wang et al. (2013); Ryzhkov et al. (2014); Thompson et al. (2015, 2018); Wang et al. (2018); Giangrande et al. (2019)), Figure 3 and Table 2 show power law fit relationships in the form of $R = aX^b$

or $R = aX^b Y^c$. The fits in Table 2 take the linear forms of $Z_h$, $Z_{dr}$, and $A_h$ as inputs, denoted as $z_h$, $z_{dr}$, and $a_h$ respectively. A bootstrap approach where 1000 fits from 10,000 randomly chosen DSDs, with replacement, from the VDIS dataset, similar to the approach taken by Wang et al. (2018), provided confidence intervals for each fit. The fit curve from each of the 1000 fits is plotted in Figure 3. The width of the 95% confidence intervals of $a$, $b$, and $c$ of each fit (Table 2) are less than 5% of the





mean $a$, $b$, and $c$ for each randomly generated fit. Furthermore, the randomly generated fits are overlapping in Figure 3, with differences in $R$ less than 10%. This therefore shows that the generated fits are robust.

## 4 Assessment of applicability of different power law retrievals

Using the 4-year Darwin dataset of DSDs, $R$, and simulated radar moment data shown in Section 3.2, this section assesses the applicability of dual polarization moments and specific attenuation for $R$ estimation in Darwin. Three criteria will be examined. First, this study estimates the spread in the p.d.f. of $R$ for given radar moments in order to estimate their potential uncertainty incurred in a power-law fit of $R$ from using these moments as input data. Secondly, a principal component analysis (PCA) is conducted to determine whether dual polarization moments or specific attenuation best contribute to the variability in $R$. This exercise is designed to guide $R$ retrieval development. Finally, the $R$ estimators developed in Section 3.2 are applied to the CPOL dataset and compared against VDIS-observed $R$ in order to test the performance of these retrievals with observed C-band radar data.

### 4.1 Uncertainty due to use of fit

One metric by which the applicability of given radar retrievals can be assessed is by examining the uncertainty, or potential for uncertainty, of the retrieval by considering the spread of $R$. While in Section 3.2 the bootstrap approach showed that the there is little difference in the power law fits due simply to random sampling, there can be an order of magnitude variability in the distribution of $R$ for a given $Z_h$, $K_{dp}$, $Z_{dr}$, or $A_h$, creating an uncertainty due to the use of a fit. This is why many studies used multiple linear regression, with multiple input variables, to form more constrained power-law $R$ estimators. To define a metric for potential uncertainty in $R$, this study calculates the spread in the p.d.f of $R$ for a given radar observable by subtracting the first quartile of $R$ from the third quartile of $R$ taken over 40 log-uniformly spaced bins of the given radar observable (Kirstetter et al., 2015). The range of these bins are 0 to 70 dBZ for $Z_h$, $10^{-3}dB\ km^{-1}$ to $100dB\ km^{-1}$ for $A_h$, $10^{-3\circ}\ km^{-1}$ to $10^{\circ}\ km^{-1}$ for $K_{dp}$, and 0 to 10 dB for $Z_{dr}$. The results of these p.d.f spread calculations as a function of the mean $R$ over each radar observable bin are shown in Figure 4 for the C- and-X band simulated radar quantities.

In Figures 4ab, the $A_h$-based estimators give the lowest spreads, followed by $K_{dp}$- then $Z_h$-based estimators for time periods when $R < 10\ mm\ hr^{-1}$. However, it is important to note that, at these $R < 10\ mm\ hr^{-1}$, the noisier nature of $K_{dp}$, and hence $A_h$ makes the applicability of these quantities to $R$ estimators questionable. In fact, numerous past studies using CSAPR and XSAPR have found the use of $K_{dp}$- and $A_h$-based estimators to be only applicable or preferable only for conditions with $Z_h > 35$-40 dBZ, present at rainfall rates greater than roughly $10\ mm\ hr^{-1}$ (Park et al., 2005b, a; Ryzhkov et al., 2005; Giangrande et al., 2014b). Algorithms by Cifelli et al. (2011) and Thompson et al. (2018) use data quality thresholds to avoid use of noisy input data in $R$ estimators. Given that Table 1 shows that the mean $Z_h$ is under 40 dBZ for $R < 10\ mm\ hr^{-1}$, this suggests that using the $Z_h$-based estimators is the most viable option when $1 < R < 10\ mm\ hr^{-1}$, similar to results using the CSU blended algorithm in Cifelli et al. (2011), Thompson et al. (2018), and Rutledge et al. (2019). However, these $Z_h$-based estimators produced the highest p.d.f. spreads and greatest $R$ uncertainty, shown in Figure 4 and also quantified by Thompson




et al. (2018). The current analysis and these prior studies highlight limitations in estimating light rainfall rates from scanning radars, since they must rely on $Z_h$ in $1 < R < 10\ mm\ hr^{-1}$.

Looking at $R > 10\ mm\ hr^{-1}$, the p.d.f.s of $R$ from $K_{dp}$-based estimators have the lowest spread at both C and X band in Figure 4. In the blended algorithm used by Cifelli et al. (2011), Thompson et al. (2018), and Rutledge et al. (2019), $K_{dp}$ is used
much more frequently for $R$ estimation at $R >> 10\ mm\ hr^{-1}$ because it exceeds necessary data quality thresholds. The $D_0$ was higher in convective DSDs with $R > 10\ mm\ hr^{-1}$ (Table 1), meaning the drop populations were more oblate, produced more total liquid water and rain, and therefore produced significant $K_{dp}$ (Bringi and Chandrasekar, 2001). For instance, in Table 1, $D_0$ increases from 1.54 mm to 1.77 mm with increasing $R$ when $R > 10\ mm\ hr^{-1}$ for the convective DSDs, also shown by Thompson et al. (2018). By definition, $K_{dp}$ becomes proportional to $W$ and $R$ once drops are large enough to be oblate (Bringi
and Chandrasekar, 2001). This is consistent with results that show $K_{dp}$ is highly correlated with $R$ when large, oblate drops are present and therefore when $R > 10\ mm\ hr^{-1}$. The spread in $R$ vs $K_{dp}$ is lower for ranges of $R > 10\ mm\ hr^{-1}$ compared to the other observables at the ARM TWP Darwin site.

Figure 4 shows that the spread in the p.d.f is lower when multiple radar observables are considered over the entire range of $R$ compared to when a single observable is used. In particular, the spread in the p.d.f. of $R$ is lowest when $K_{dp}$ and $Z_{dr}$ are
constrained for time periods when $R > 1.5\ mm\ hr^{-1}$. Even the use of $Z_h$, and $Z_{dr}$ as constraints lowers the spread in the p.d.f. of $R$ compared to using a single radar observable. $(K_{dp}, Z_{dr})$-based estimators are used to estimate $R$ when $R > 10\ mm\ hr^{-1}$ in the blended algorithm used by Cifelli et al. (2011), Thompson et al. (2018), and Rutledge et al. (2019). Therefore, similar to past studies in Colorado, Oklahoma, and Manus and Gan Island, this shows that using multiple linear regression reduces the uncertainty in $R$ due to the use of fits for the CPOL data in Darwin.

**4.2  Principal component analysis**

The previous subsection revealed which rainfall rate estimators most minimize the spread in the p.d.f. of $R$ for these Darwin datasets. Now, this section explores the utility of using $Z_h$, $Z_{dr}$, $K_{dp}$, or $A_h$ to estimate $R$. To do this, this section shows a PCA conducted on the simulated VDIS radar moments and $A_h$. The first two components explained over 99% of the variance in $R$. The PCA was applied to $A_h$, $Z_h$, and $Z_{dr}$, in order to minimize variability due to the many orders of magnitude that
these variables span. The results of this PCA are shown in Figure 5 for C-band and 6 for X-band. The explanatory power of the second principal component was small compared to the first, only becoming non-negligible at $R > 100\ mm\ hr^{-1}$ in Figure 5 and at $R > 10\ mm\ hr^{-1}$ Figure 6.

The previous analysis in Table 1 and Figure 2 shows that stratiform and possible weak convective clouds primarily contribute to rainfall for times when $R < 10\ mm\ hr^{-1}$ while stronger convective rain classified by BR98 had higher $R$. Since we expect
rainfall from stratiform and convective clouds to have DSDs with different characteristics for a given $R$, and $R = 10\ mm\ hr^{-1}$ was a suitable threshold to distinguish deep convection from weaker convection and stratiform rain, the PCA in Figure 5 and 6 is further stratified by $R$ in order to account for this DSD variability. When restricting the PCA to $1 < R < 10\ mm\ hr^{-1}$, only $Z_h$ explains any of the variance of $R$ in Figure 5 and 6. This is again consistent with past efforts that have preferred $Z_h$





estimators for estimating these lighter rainfall rates from scanning radars (Cifelli et al., 2011; Park et al., 2005b; Ryzhkov et al., 2005; Giangrande et al., 2014a, 2019; Thompson et al., 2018; Wang et al., 2018; Rutledge et al., 2019).

The second principal component has high absolute values in the $K_{dp}$ and $Z_{dr}$ directions at $10 < R < 100 \ mm \ hr^{-1}$. The variance due to the second principal component with higher absolute values in the $K_{dp}$ and $Z_{dr}$ directions starts to exceed 1%,

for $R > 100 \ mm \ hr^{-1}$. Thus, $K_{dp}$ and $Z_{dr}$ become better suited for $R$ estimation as $R$ increases, which confirms prior studies (e.g. Bringi and Chandrasekar (2001)). The first principal component in the $K_{dp}$ direction is 0.25 at $1 < R < 10 \ mm \ hr^{-1}$, and 0.7 at $R > 100 \ mm \ hr^{-1}$. Therefore, it is clear from this analysis that $K_{dp}$ has more predictive power for $R$ at these higher rainfall rates. This further confirms and quantifies prior studies that showed $K_{dp}$-based estimators for $R$ are successful at higher rain rates (Sachidananda and Zrnic', 1985; Sachidananda and Zrnić, 1987). This is also consistent with previous

studies recommending the use of $K_{dp}$-based estimators over $A_h$-based estimators for CSAPR and XSAPR at the ARM SGP site to sample deep convection (Giangrande et al., 2014b).

The PCA shows that $A_h$ has little predictive capability for $R$ beyond the other observables for DSDs in Darwin except for $R > 100 \ mm \ hr^{-1}$ at X-band in Figure 6a. Past ARM efforts at SGP have shown that, for XSAPR radars, severe attenuation prohibited accurate rainfall estimation for this range of $R$ (Giangrande et al., 2014b). So, while $A_h$ may provide more predictive

capability at X-band at $R > 100 \ mm \ hr^{-1}$, it is unlikely that XSAPRs could utilize $A_h$ in these conditions for practice. Therefore, this and the PCA shows that using $A_h$ for developing $R$ estimators for Darwin ARM radars and CPOL, in coastal tropical regions provides no advantage over using other radar observables.

## 5   Comparisons of CPOL retrievals with VDIS

As a final metric for evaluating the applicability of various radar quantities to the development of $R$ estimators for ARM

radars deployed at the TWP ARM site, the $R$ estimators for C-band radars in Section 2.2 were applied to the gate immediately over the VDIS at 0.51 km altitude above ground level. First, in order to compare the estimators, Figure 7 shows scatter plots of $R$ observed from VDIS compared to observed $Z_h$, $K_{dp}$ and $A_h$ from CPOL with the $R$ estimator developed from the VDIS data overlaid on the scatter plot. It is apparent that there is one-to-two orders of magnitude of scatter in $R$, and ($RMSE > 8.5 \ mm \ hr^{-1}$), in all panels of Figure 7. For Figures 7ac, the $Z_h$ values are generally lower than the fit- produced

values for $R > 50 \ mm \ hr^{-1}$. While the $Z_h$ are, to the best of possible efforts, adjusted for attenuation, there is still the possibility that $Z_h$ remains affected by attenuation at these high $R$ that was uncorrectable. In addition, factors including the horizontal advection and breakup of drops as they travel from the CPOL sample gate to the VDIS, as well as noise in the $K_{dp}$ and $A_h$ fields at $R < 10 \ mm \ hr^{-1}$, are likely inducing scatter. Figure 7 therefore shows that a single radar observable does not adequately describe the full variability of $R$. As shown in many prior studies, $R$ estimation could be improved by

applying a blend of $R$ estimators depending on rain conditions or radar multivariable conditions, or by employing multiple linear regression with more variables in each $R$ estimator.

In order to determine the blend, or set of $R$ estimators based on rain conditions, that can provide the best agreement with VDIS observations, Figure 8 shows comparisons of 10-minute averages of $R$ estimated from CPOL $Z_h$, $A_h$, and $K_{dp}$. The





input data to the $R$ estimators are from the CPOL gate immediately over VDIS. The rain-branch of the CSU-blended technique, originally developed for Colorado by Cifelli et al. (2011), and then subsequently modified for the tropical oceans by Thompson et al. (2018), is also included in Figure 8. The CSU-blended technique uses a decision tree based on data quality thresholds for $Z_h$, $K_{dp}$, and $Z_{dr}$ to select an $R$ estimator. The modifications by Thompson et al. (2018) were developed for conditions

in Manus and Gan Island where stronger convection that produces hail that can further melt into large surface raindrops is not common. Therefore, since such convection is common in Darwin, the estimators used by the CSU-blended technique to generate Figure 8 were changed to those in Table 2 in order to more accurately represent the local DSDs sampled in Darwin over several years. The data quality thresholds used by (Thompson et al., 2018) for tropical oceans were also used here, so only the coefficients of the $R$ estimators were changed.

The analysis, here, and in previous studies shows that different microphysical processes likely occur at different ranges of $R$. Namely, raindrops forming from melting hail are unlikely at $1 < R < 10 \ mm \ hr^{-1}$, but more likely during times when $R > 10 \ mm \ hr^{-1}$. Therefore, in order to analyze how the agreement between estimated $R$ and VDIS-observed $R$ changes for these different conditions, the mean, 5th, and 95th percentiles of the CPOL estimated $R$ for log-uniformly spaced intervals of VDIS-observed $R$ are shown in Figure 8. The first focus is on time periods with $1 < R < 10 \ mm \ hr^{-1}$ where stratiform rain

and weaker convection are more likely present. It is clear that the mean $R$ estimated by the $K_{dp}$-, $(K_{dp},Z_{dr})$- and $A_h$-based estimators greatly overestimate the mean VDIS observed $R$ by over a factor of 10 for these conditions (Figure 8bce). However, the mean $R$ from both the CSU-blended technique and the $Z_h$-based estimators are on average 12% higher than the mean VDIS-observed $R$ (Figure 8af). Consistent with the previous analysis, this again supports the notion that $Z_h$-based estimators, and the modified CSU-blended technique, are most appropriate for use in these conditions characterized by stratiform and

weak convective rainfall.

     Switching focus to analyzing conditions of $R > 10 \ mm \ hr^{-1}$, in which strong convection that is capable of forming hail that melts into raindrops is much more likely, $R$ from CPOL calculated from $Z_h$- and $(Z_h,Z_{dr})$-based estimators underestimate the VDIS-observed $R$ (Figure 7ae). On average, the mean estimated $R$ from the modified CSU-blended technique is 17% lower than the mean observed $R$ (Figure 8f), while the $K_{dp}$-estimated $R$ is 21% higher (Figure 8c), $A_r$-estimated $R$ is 22%

higher (Figure 8d), and $K_{dp}, Z_{dr}$-estimated $R$ is 36% higher (Figure 8d) for these time periods. This demonstrates that $R$ estimated from the CSU-blended technique, on average, provides the best agreement with VDIS-observed $R$ for these time periods dominated by stronger convection. The CSU-blended technique also gave estimates of $R$ in best agreement with VDIS-observed $R$ for the time periods dominated by stratiform rain and weaker convection. Therefore, this demonstrates that the use of CSU-blended technique, with modifications to the coefficients of the $R$ estimators for Darwin DSDs, provides the

optimal estimate of $R$ for the CPOL data in Darwin.

## 6   Conclusions

The C-band POLarization (CPOL) Radar at the U.S. Department of Energy Atmospheric Radiation Measurement (ARM) Tropical Western Pacific (TWP) site in Darwin has been operating for over a decade and thus provides an ample dataset for



developing essential rainfall climatologies. These are important for understanding rainfall variability and for validation of global climate models. A crucial quantity in this dataset includes the rainfall rate $R$. $R$ is not detected directly by radars, but is retrieved from radar observables such as radar reflectivity factor $Z_h$, differential reflectivity $Z_{dr}$, specific differential phase $K_{dp}$, and specific horizontal attenuation $A_h$. Algorithms using $A_h$ at S-band have been successful for the NOAA NEXRAD radars,

but studies utilizing ARM C- and X-band scanning radars at the ARM Southern Great Plains site have shown that retrievals using $K_{dp}$ without $Z_{dr}$ are most successful. Most prior studies are based off of limited data compared to the four year dataset available from ARM TWP Darwin site. This therefore motivated a study to determine which of these radar observables are most applicable for retrieving rainfall retrievals for the CPOL and ARM radars in Darwin. We use a much larger dataset than previous efforts. We first developed $R$ estimators from simulated $Z_h$, $Z_{dr}$, $K_{dp}$, and $A_h$ from video disdrometer (VDIS) data

in Darwin for C- and X-band radar wavelengths. The VDIS observations generally showed that Darwin rainfall is typically stratiform (in terms of frequency), having median drop diameters $D_0$ less than 1.5 mm at $R < 10\ mm\ hr^{-1}$. Rainfall here contributed by convection had $D_0 > 1.5\ mm$ for $R > 10\ mm\ hr^{-1}$, consistent with past observations in Darwin.

In order to determine the applicability of $Z_h$, $Z_{dr}$, $K_{dp}$, and $A_h$ a principal component analysis (PCA) was conducted on $Z_h$, $Z_{dr}$, $K_{dp}$, and $A_h$ simulated from the VDIS DSDs. The majority of the variability in $R$ was attributable to variations in $Z_h$

for $1 < R < 10\ mm\ hr^{-1}$, with $K_{dp}$ becoming increasingly important for explaining $R$ variability at $R > 10\ mm\ hr^{-1}$. $A_h$ provided little additional predictive capability beyond the other observables for $R$ observed by the VDIS for convection over Darwin. The uncertainty in $R$ estimators fitted to these radar inputs was estimated. It was shown that the use of $K_{dp}$-based estimators, as well as estimators based on multiple observables, minimizes this uncertainty, similar to results from prior studies.

To further assess the applicability of the various radar observables to estimating $R$, these different $R$ estimators (formed

from simulated VDIS radar variables) were tested on CPOL observations at the radar gate about 0.56 km over the VDIS. Each $R$ estimator was tested individually, and also in the rain-based branch of the CSU Blended algorithm that chooses between estimators based on data quality thresholds (Cifelli et al., 2011; Thompson et al., 2018; Rutledge et al., 2019). The highest-performing estimation techniques for X- and C- band at Darwin were the CSU-blended technique for rainfall rates of $1 < R < 10\ mm\ hr^{-1}$ as well as when $R > 10\ mm\ hr^{-1}$. This demonstrates that the CSU-blended technique is best for

stratiform and weak convective rain as well as strong convection in Darwin. Local $R$ estimators were used in the blended algorithm to more accurately represent DSDs in Darwin. The methodology used in this study could be used in future studies to quantify uncertainty in $R$ estimation methods.

*Code and data availability.* The code used for the analysis of the CPOL data is available at http://www.github.com/EVS-ATMOS/cmdv-rrm-anl/. The CPOL data can be downloaded from the Atmospheric Radiation Measurement Facility archive.



*Author contributions.* RJ conducted most of the data analysis and contributed greatly to the writing of the manuscript. SC, ET, BD, SG, and DW contributed to many of the research ideas and writing in this manuscript. VL and AP processed the CPOL data and provided feedback on the writing of the manuscript. SP provided provided feedback on the writing of the manuscript.

*Competing interests.* There are no competing interests present in this manuscript.

*Acknowledgements.* Argonne National Laboratory's work was supported by the U.S. Department of Energy, Office of Science, Office of Biological and Environmental Research, under Contract DE-AC02-06CH11357. This work has been supported by the Office of Biological and Environmental Research (OBER) of the U.S. Department of Energy (DOE) as part of the Climate Model Development and Validation activity. NOAA PSL contributes effort with funding from the Weather Program Office's Precipitation Prediction Grand Challenge. The development of the Python ARM radar toolkit, was funded by the ARM program part of the Office of Biological and Environmental Research
(OBER) of the U.S. Department of Energy (DOE). The work from Monash University and the Bureau of Meteorology was partly supported by the U.S. Department of Energy Atmospheric Systems Research Program through the grant DE-SC0014063. We gratefully acknowledge use of the Bebop cluster in the Laboratory Computing Resource Center at Argonne National Laboratory. The bulk of the code has been written using the open-source NumPy, Scipy, Matplotlib, Jupyter and Dask projects, and the authors are grateful to the authors of these projects. BDs contributions are supported by the U.S. Department of Energy Atmospheric Systems Research Program through the grant
DE-SC0017977.





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



**Table 1.** Mean DSD parameters for given $R$ ranges and BR09 convective-stratiform classification.

| R range [$mm\ hr^{-1}$] | No DSD | | Mean R [$mm\ hr^{-1}$] | | $\log_{10} N_w$ [$m^{-3}\ mm^{-1}$] | | $D_0$ [mm] | | $Z_h$ [dBZ] | | $W$ [$g\ m^{-3}$] | |
|---|---|---|---|---|---|---|---|---|---|---|---|---|
| | C | S | C | S | C | S | C | S | C | S | C | S |
| 0.5-2 | 679 | 15052 | 1.06 | 1.07 | 2.40 | 1.17 | 2.29 | 3.38 | 4.0 | 24.8 | 0.06 | 0.05 |
| 2-4 | 443 | 6568 | 2.97 | 2.85 | 3.08 | 3.66 | 1.96 | 1.26 | 32.1 | 30.5 | 0.14 | 0.13 |
| 4-6 | 436 | 2694 | 5.00 | 4.84 | 3.38 | 3.89 | 1.79 | 1.27 | 34.8 | 33.2 | 0.24 | 0.23 |
| 6-10 | 1216 | 1536 | 8.08 | 7.36 | 3.85 | 4.13 | 1.54 | 1.23 | 36.3 | 35.2 | 0.39 | 0.36 |
| 10-20 | 2422 | 257 | 14.3 | 12.6 | 4.07 | 4.34 | 1.52 | 1.24 | 39.5 | 39.7 | 0.69 | 0.60 |
| 20-40 | 2036 | 24 | 28.5 | 25.4 | 4.19 | 4.53 | 1.66 | 1.32 | 44.2 | 48.3 | 1.33 | 1.24 |
| 40-60 | 956 | 0 | 48.9 | n/a | 4.32 | n/a | 1.75 | n/a | 48.1 | n/a | 2.30 | n/a |
| 60+ | 887 | 0 | 85.9 | n/a | 4.54 | n/a | 1.77 | n/a | 51.5 | n/a | 4.11 | n/a |





**Table 2.** 95% confidence intervals of the generated fit parameters, RMSE, and correlation coefficient for each fit.

| Relationship | a | b | c | RMSE $[mm\ hr^{-1}]$ | Correlation coefficient |
|---|---|---|---|---|---|
| **C-band** | | | | | |
| $R(z_h)$ | $0.0208 \pm 0.00$ | $0.70 \pm 0.0002$ | | 16.66 | 0.95 |
| $R(z_h)$ convective | $0.0520 \pm 0.00002$ | $0.62 \pm 0.0001$ | | 35.30 | 0.91 |
| $R(z_h)$ stratiform | $0.0307 \pm 0.00005$ | $0.63 \pm 0.0002$ | | 2.75 | 0.90 |
| $R(K_{dp})$ | $26.116 \pm 0.01$ | $0.77 \pm 0.0001$ | | 4.67 | 0.96 |
| $R(a_h)$ | $258.89 \pm 0.31$ | $0.87 \pm 0.0002$ | | 12.76 | 0.79 |
| $R(z_h, z_{dr})$ | $0.0122 \pm 0.0000$ | $0.85 \pm 0.0002$ | $-4.25 \pm 0.006$ | 4.09 | 0.97 |
| $R(K_{dp}, z_{dr})$ | $46.2347 \pm 0.06$ | $0.90 \pm 0.0002$ | $-1.69 \pm 0.006$ | 3.86 | 0.99 |
| **X-band** | | | | | |
| $R(z_h)$ | $0.0207 \pm 0.00004$ | $0.66 \pm 0.000$ | | 16.03 | 0.94 |
| $R(z_h)$ convective | $0.0853 \pm 0.00003$ | $0.56 \pm 0.000$ | | 35.2 | 0.88 |
| $R(z_h)$ stratiform | $0.0360 \pm 0.00006$ | $0.61 \pm 0.000$ | | 2.79 | 0.89 |
| $R(K_{dp})$ | $17.5432 \pm 0.006$ | $0.77 \pm 0.0001$ | | 4.38 | 0.97 |
| $R(a_h)$ | $49.54 \pm 0.027$ | $0.80 \pm 0.0001$ | | 5.56 | 0.79 |
| $R(z_h, z_{dr})$ | $0.011 \pm 0.00001$ | $0.89 \pm 0.0001$ | $-5.35 \pm 0.002$ | 4.20 | 0.98 |
| $R(K_{dp}, z_{dr})$ | $31.794 \pm 0.017$ | $0.94 \pm 0.0001$ | $-1.99 \pm 0.002$ | 3.88 | 0.98 |



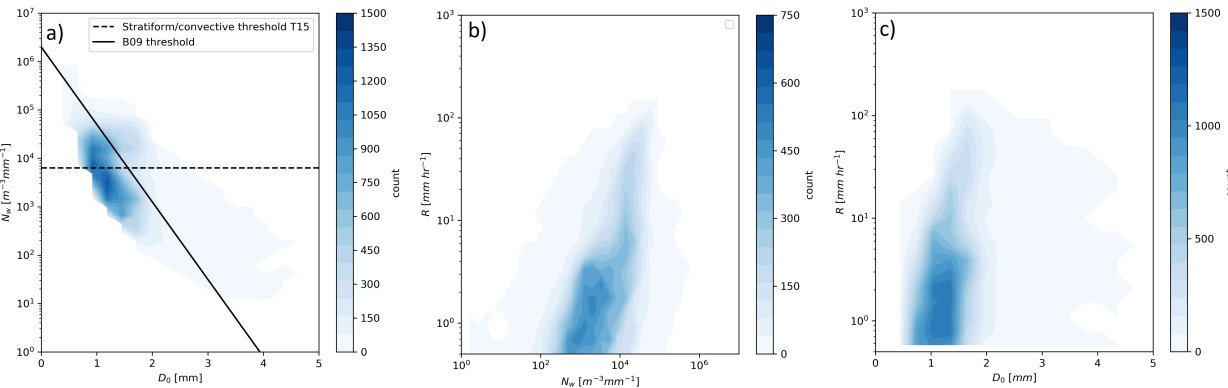

**Figure 1.** Frequency histogram of **(a)** $N_w$ and $D_0$, **(b)** $R$ and $D_0$, and **(c)** $R$ and $N_w$ from the VDIS for all of the DSDs containing more than 100 drops in each sample. The criteria used to classify stratiform and convective DSDs from Bringi et al. (2009) (BR09) and Thompson et al. (2015) (T15) are shown by the lines.



**Figure 2.** Normalized frequency histograms of **(a)** $N_w$, **(b)** $D_0$, **(c)** $W$, and **(d)** $R$ separated by the BR09 stratiform-convective classification.

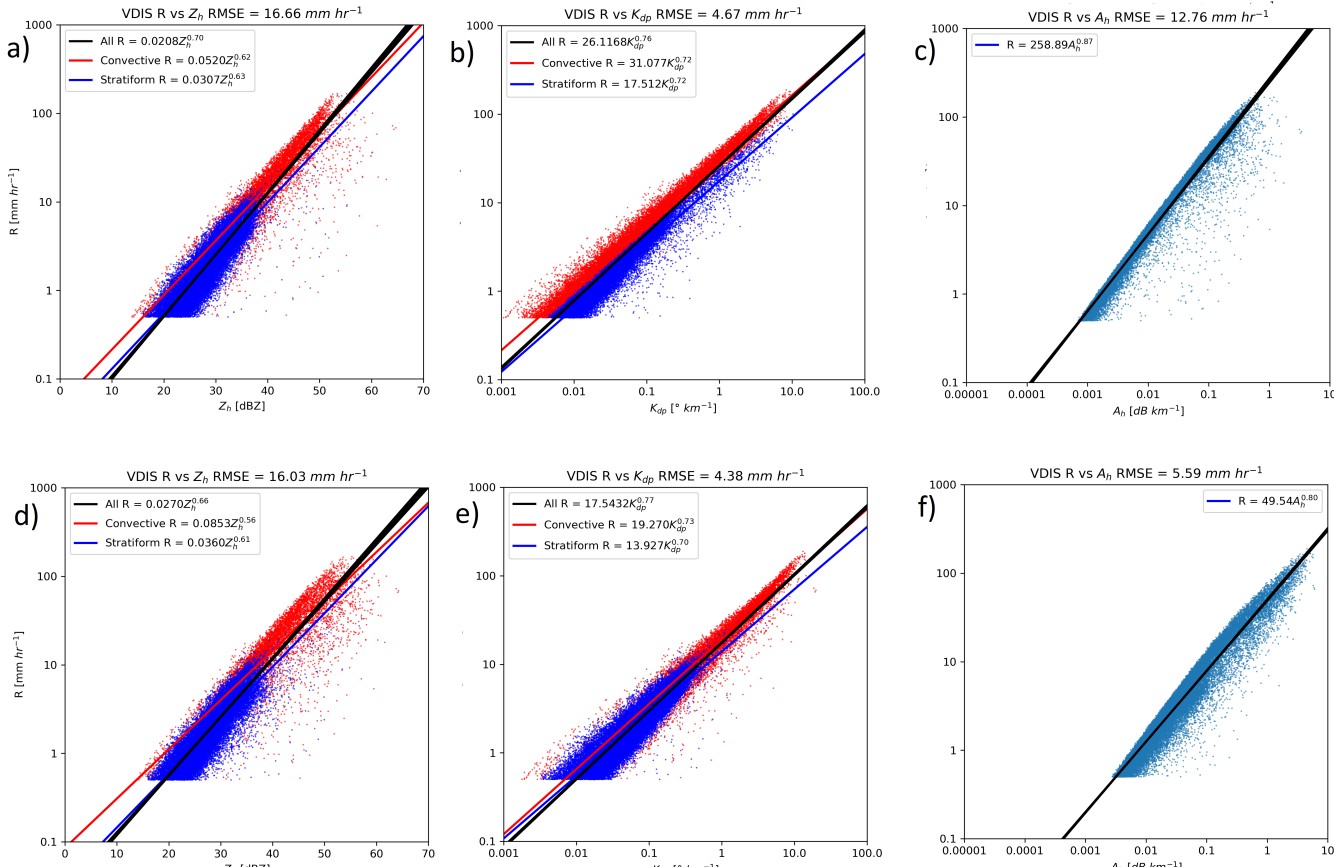

**Figure 3.** $R$ from VDIS as a function of **(a)** $Z_h$, **(b)** $A_h$, **(c)** $K_{dp}$, for the simulated radar moments from VDIS at C-band. **(d-f)** as in **(a-c)** but for X-band. Each colored line represents a power law best fit of the variables. Each black line represents a fit produced by the bootstrap technique applied to the data in each panel.



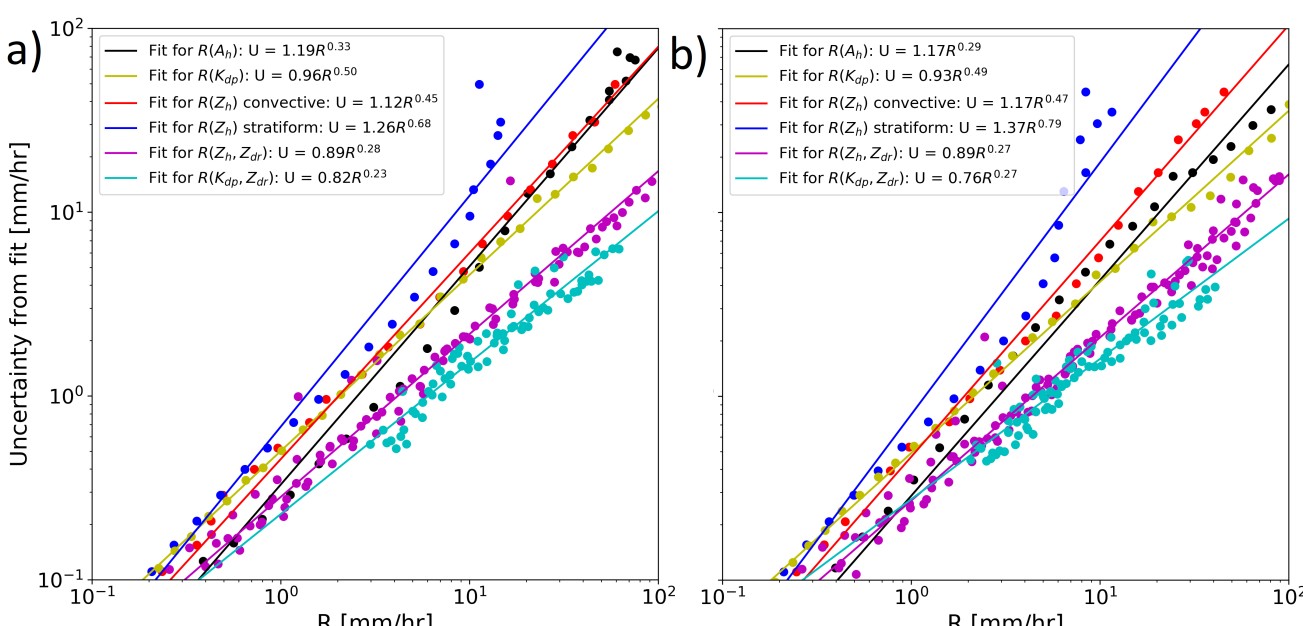

**Figure 4.** The spread in the p.d.f. of $R$ estimated using the methodology of Kirstetter et al. (2015) as a function of mean $R$ for given ranges of radar moments at C-band **(a)**, and X-band **(b)**

.



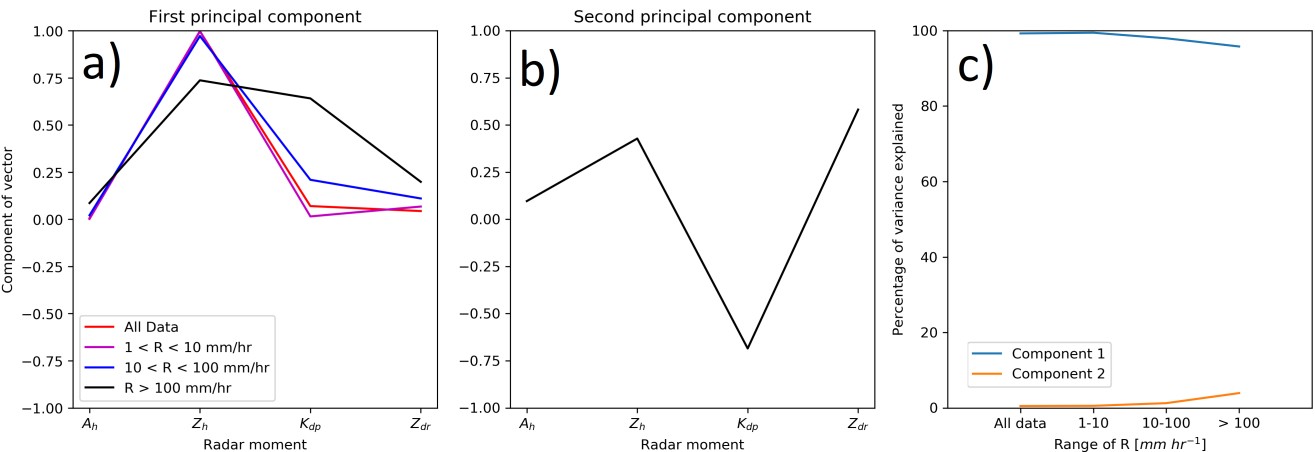

**Figure 5.** The first **(a)** and second **(b)** principal components of the simulated moments in $(A_h, Z_h, K_{dp}, Z_{dr})$ phase space for S-band. **(c)** shows the variance in $R$ explained by each principal component for the C-band simulated moments.





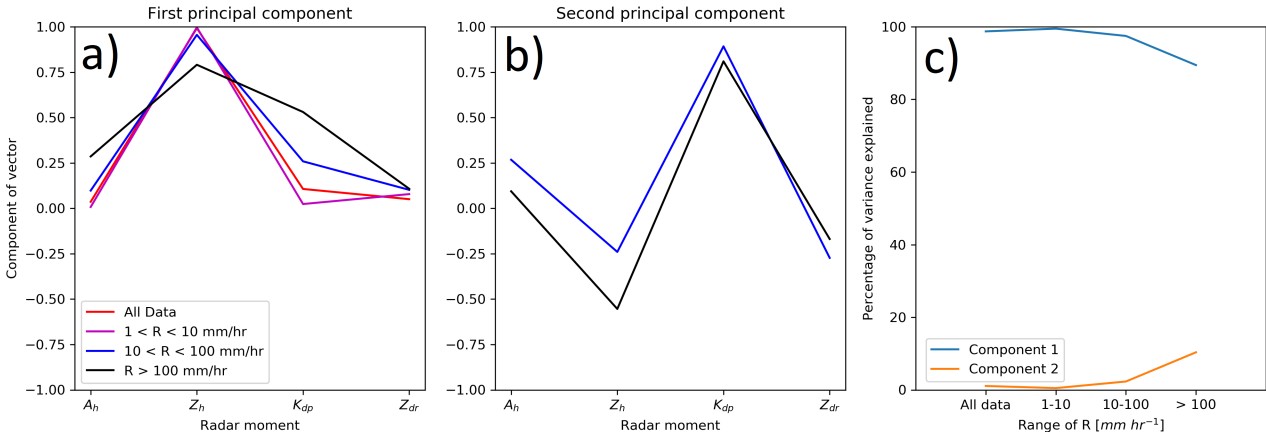

**Figure 6.** As Figure 5 but for X-band.



**Figure 7.** $R$ from the VDIS as a function of **(a)** $Z_h$ for convective DSDs, **(b)** $K_{dp}$, **(c)** $Z_h$ for stratiform DSDs, and **(d)** $A_h$ from CPOL. Solid lines are the $R$ estimators in Figure 3.

**Figure 8.** Normalized frequency distributions 10 minute averages of $R$ estimated from the lowest gate from CPOL over VDIS using given estimators in Figure 3 as a function of 10 minute averages of $R$ recorded by the VDIS. Estimators used to estimate $R$ from CPOL are shown in each panel. Solid lines denote medians, dashed lines are the 5th and 95th percentiles of the estimated $R$ from CPOL.