# Peer review of "The development of rainfall retrievals from radar at Darwin"

_Atmospheric Measurement Techniques, 2020_

## Referee Comment (RC1) · Anonymous Referee #1 · 3 Aug 2020

August 3, 2020

**General comments**

The manuscript presents a statistical analysis of polarimetric radar data and in-situ (disdrometer) observations collected over several years in Australia. The topic is relevant for the readership of AMT and the availability of such dataset is very important for future studies. I appreciate that data (and code) are made available. My concerns about this study are related to its objective, in my view not clearly defined, and to the methodological aspects that should be more thoroughly presented. In particular, I expect to see a critical discussion of a few crucial points (comparison of point-measurements with radar volumes, beam broadening effects, gridding of polar data, see the points listed below). I overall recommend a major revision of the manuscript.

1. The stated objective of this manuscript is to provide (improved) information for validation of global circulation models. However, i do not see any tentative in this direction. The research presented here is a statistical analysis of disdrometer data and radar data in a tropical region. The goal is therefore not clearly defined and it needs some rephrasing / revision.

2. Should also the distance with respect to the radar be taken into account in your

analysis? With in mind attenuation, beam broadening, partial beam filling, it is an important parameter. See also specific comment 3 below.

3. Section 2.2: i cannot understand a few things in this section. How many dis-drometers are used? Where is/are the video-disdrometer(s?) located (a map will help)? . Are they co-located with the radar? Are they distributed in a net-work? According to which strategy and which assumptions are volume-based radar measurements compared with point-based disdrometer measurements? Please adapt this section in order to provide the necessary (and very important) information: it is difficult to provide a useful review and relevant suggestions when this aspect is not clear.

**Specific comments**

1. P1: CSU, VDIS: undefined acronyms.

2. P2, L1: I would mention also the other fields where accurate rainfall estimation is crucial: nowcasting, alert issuing, climatology (etc.).

3. P2, L21: I find this part slightly over-simplified. Beam broadening should be discussed and explained. While the radial resolution can be of 100m in polar coordinates, this will not be true in Cartesian coordinates when we are far from the radar. The conversion from polar data (radar) to Cartesian grid data seems to me a crucial point to discuss, especially given the goal of the manuscript: provide knowledge useful for comparison with global circulation models.

4. P2, L35: The other side of the medal of $Z_{DR}$ and $K_{dp}$ is that: (1) they use two channels, so there is twice the possibility that an hardware issue will affect them, (2) $K_{dp}$ is not a radar observable, but it needs to be estimated from $\Psi_{dp}$: (Otto

and Russchenberg, 2011; Wang and Chandrasekar, 2009; Grazioli et al., 2014), (3) $Z_{DR}$ is affected by differential attenuation and it is affected by the incidence angle (Ryzhkov et al., 2005)

5. P4, L16-20: how are the various elevations combined to provide a proxy of precipitation near ground level?

6. Section 4: i had from time to time some difficulties to understand where the polarimetric variables used in this section where coming from (i.e., simulated from VDIS or measured from the radar). I suggest to clarify this aspect through the manuscript, and maybe use a different notation for simulated variables (like $Z_H^*$).

7. P12: data availability. Please provide also the link for the data archive.

8. $K_{dp}$: a few words about the estimation accuracy of this parameter should be provided.

**References**

T. Otto and H. W. J. Russchenberg. Estimation of specific differential phase and differential backscatter phase from polarimetric weather radar measurements of rain. *IEEE Geosci. Remote Sens. Lett.*, 8(5):988–992, 2011. doi: 10.1109/LGRS.2011.2145354.

Y. T. Wang and V. Chandrasekar. Algorithm for estimation of the specific differential phase. *J. Atmos. Oceanic Technol.*, 26(12):2565–2578, 2009. doi: 10.1175/2009JTECHA1358.1.

J. Grazioli, M. Schneebeli, and A. Berne. Accuracy of phase-based algorithms for the estimation of the specific differential phase shift using simulated polarimetric weather radar data. *IEEE Geosci. Remote Sens. Lett.*, 11(4):763–767, 2014. doi: 10.1109/LGRS.2013.2278620.

A. V. Ryzhkov, S. E. Giangrande, V. M. Melnikov, and T. J. Schuur. Calibration issues of dual-polarization radar measurements. *J. Atmos. Oceanic Technol.*, 22(8):1138–1155, 2005. doi: 10.1175/JTECH1772.1.

---

## Referee Comment (RC3) · Anonymous Referee #2 · 3 Aug 2020

GENERAL COMMENT

This paper presents an analysis of the applicability of dual-polarization rainfall relations for C-band radars. Although in general well written, with adequate reference to previous works and (mostly) clear illustrations, I found the approach presents some flaws, specifically:

- Simulations (section 3.2): the measurement uncertainty is not considered in these simulations. Therefore, these results only show the parametric error. For actual applications, the measurement errors should be included in the simulations. For example, a two-parameter relation like R(Zh,Zdr) has lower parametric error than R(Zh), but may have a larger total error depending on the measurement accuracy of Zdr. Have the actual measurement errors of CPOL been considered somehow?

- PCA analysis (section 4.2): this technique is used in this paper as an original contribution for application to dual-polarization radar rainfall estimation. The physical meaning of the results is not always clear. For example, in fig. 5 (panel b) it is not clear if all three lines are the same or something is missing. In panel a) it is a bit confusing to see the first component of Ah close to 0, after having seen an excellent correlation in fig. 3... Although I recognize that this can be related to a lack of familiarity with PCA analysis, I encourage to authors to provide more details about the analysis performed and better discussion of the results presented in fig. 5 and 6.

I encourage the authors to revise the manuscript, in particular the simulation and PCA analysis sections. Also, the three parts (simulation, PCA analysis, comparison with disdrometer) are treated quite independently and there is little comprehensive discussion in the final section. I would expect in the Conclusions a more in-depth discussion of the key findings and eventually contrasting results obtained with the different methods. As a specific example, I found the conclusion about Ah (it is said that it has little predictive capability) not enough supported by compelling arguments, nor it is considered the fact that several estimators exist for the estimation of Ah (and for Kdp) with quite different behavior.

SPECIFIC COMMENTS AND MINOR CORRECTIONS

- Units should be in Roman font (not Italic), e.g. mm/h.

- P2, L4: "...magnitude OF the diurnal cycle.."

- P3, L25: ".. were developed and using data.." change to ".. were developed using data.."?

- P4, L28-29: "In addition, Zh and Zdr at C-band are prone to (differential) attenuation from heavy rainfall which may bias (underestimate) R". This sentence needs to be reformulated because underestimation of Zdr causes overestimation (not underestimation) of R. In the use of R(Zh,Zdr) estimator, the underestimation of Zh and Zdr due to

attenuation tend to (at least partially) compensate because of the opposite sign of the exponents.

- P4, L32: linear programming is used to estimate Kdp. More discussion on this specific estimation method may be needed, especially considering plots like in fig. 8: may the positive biased estimates R(Kdp) at low rain rates may be attributed to the specific behavior of the linear programming algorithm which always produces nonnegative Kdp values?

- P5, L3: "Waldovel" -> "Waldvogel"

- P6, L7: "Darwin Colorado"?? Should it read "Darwin (Australia)"?

- P8, L6: normally "PDF" should read better than "p.d.f."

- P10, L21: What is the distance between the radar and the VDIS? A map may be useful. It is mentioned that measurements may be affected by attenuation, so it is important to know the range from the radar.

- P12, L6: "based off of limited.." -> "based on"?

- P12, L7: "retrieving rainfall retrievals". May read better: "retrieving rainfall estimates".

- P21, fig.1: panels b) and c) swapped

- P25, fig.5: replace "S-band" with "C-band"?

---

## Referee Comment (RC4) · Anonymous Referee #3 · 4 Aug 2020

General Comments

This manuscript presents a study of polarimetric relationships to estimate rainfall rates at Darwin, Australia, based on radar retrievals from the C-band dual-polarization radar (CPOL). The retrieved relationships are validated against a co-located two dimensional video disdometer (VDIS) via statistical metrics. I find the manuscript relevant for publication in AMT but I suggest the authors to perform major revisions related mainly to the organization of speech and clarity of explanation throughout the manuscript.

1) For instance, in the Introduction, many of the sentences need a fundamental reorganization (more specifics are presented in the Technical Comments). The overall meaning is understandable but often words are missing or displaced and the flow of the speech is negatively affected.

2) It is mentioned in the manuscript that the VDIS is co-located with the CPOL radar but no picture of the area where the two sensors are located nor their coordinates are provided by the authors. I suggest to clarify this and include a picture of the filed of study in Darwin.

3) I have some issues with the description of a couple of Figures. In paragraph 3.1 is described Figure 1. A mention is missing of the dashed and solid lines that separate two types of precipitation, even though the selection criteria for convective-stratiform precipitation are described later. Similar remarks apply to Figure 3 and its description in paragraph 3.2. It would be good for a better reader understanding to mention which curve (and in which panel) represents the data fit.

4) In Section 4.1, it is stated that the Ah-based estimators give the lowest spread for R<10mm/hr and Kdp-based estimators give the lowest spread for R>10mm/hr. This is true only if a combination of different moments is not taken into account but this is not mentioned in the text.

Specific Comments/Technical Corrections

1) Page 1, line 17. Please define 'VDIS'.

2) Page 2, line 4-5. Please reformulate: 'is that the phase and magnitude the diurnal cycle of precipitation are not adequately resolved due to the parameterization of convection'.

3) Page 2, line 12-14. Please reformulate the sentence.

4) Page 2, line 15. 'Similar summary'?

5) Page 2, line 20. Please replace 'this' with 'a'.

6) Page 2, line 22. Please define 'R' in the Introduction.

7) Page 3, line 1. Please replace 'found in the limits of' with 'in'.

8) Page 3, line 5. Please replace with 'with DSD observations subject to comparable limitations'.

9) Page 3, line 7. Please insert 'the' before 'aforementioned'.

10) Page 3, line 12. Please define 'DOE ARM'.

11) Page 3, line 19. Please define 'MC3E'.

12) Page 3, line 24. Please define 'RMSE'.

13) Page 3, line 25. Please remove 'and' before 'using'.

14) Page 3, line 32-33. I don't understand this sentence. Do you want to state the aim of this research? What are the challenges?

15) Page 3, line 34. 'Efforts' don't have access to data.

16) Page 4, line 7. Please replace 'for' with 'in'.

17) Page 4, line 7. No need to say 'rainfall rate R'. Either 'rainfall rate' or 'R' but make sure you defined 'R' previously.

18) Page 4, line 7. Please use present tense: 'This study uses'.

19) Page 4, line 11. 'on these quantities' is superfluous.

20) Page 4, line 21. Please remove 'a' before '250 m' and '1°'.

21) Page 4, line 30. Please define 'Z-PHI method' and include a reference.

22) Page 5, line 4. Are JW disdrometers less optimal for assessing dual-polarization radar efforts in lighter rain and/or small-drop conditions than? And, once again, 'efforts' are not a measurable variable. The authors mean 'radar moments' or 'meteorological quantities'.

23) Page 5, line 16. Please rephrase as 'After the application of these thresholds'.

24) Page 5, line 21. Incorrect reference to Wang et al. (2018) and Giangrande et al. (2019).

25) Page 6, line 3. I would not use the verb 'stratify' in a sentence where 'stratiform' is referred to the type of precipitation. Please use 'separate' or 'divide'.

26) Page 6, line 13-14. 'other datasets around the world' is too generic.

27) Page 6, line 17-18. Please finish the sentence starting with 'Prior studies find that tropical-oceanic cloud behaviors do not solely drive most of the surface rainfall here'. They drive what else then?

28) Page 6, line 26. Please define 'W'.

29) Page 6, line 34. The authors mean 'T15' here.

30) Page 7, line 16. Please add ',' after 'Therefore'.

31) Page 7, line 32-33. The sentence starting with 'A bootstrap' is a bit convoluted and needs rephrasing.

32) Page 7, line 33 – page 8, line 1. 'The width of the 95% confidence intervals of a, b, and c of each fit (Table 2) are less than 5% of the mean a, b, and c for each randomly generated fit'. Please mention some numbers in the text for a better understanding.

33) Page 8, line 6. Please define 'p.d.f.'.

34) Page 8, line 6,7,9. Be consistent with the use adverbs: 'firstly', 'secondly', and then 'finally'.

35) Page 8, line 26. Please remove one instance of 'only'.

36) Page 8, line 29. Please remove 'use of'.

37) Page 9, line 28. Please replace 'possible' with 'possibly'.

38) Page 10, line 3. It is valid for X-band radar only based on the figure.

39) Page 10, line 23. I suggest to mention, for clarity, that the estimator is represented by the dashed line.

40) Page 11, line 8. Incorrect reference to 'Thompson et al. (2018)'.

41) Page 11, line 10. Please insert a coma after 'previous studies' instead of after 'here'.

42) Page 12, line 4. 'Algorithms using Ah' for what?

43) Page 12, line 6. Please replace 'based off of' with 'based on'.

44) Page 12, line 7. 'Therefore' between comas.

45) Page 12, line 8. Please use the past tense here: we 'used' instead of 'use'. Check through the manuscript that the tenses are consistent.

46) Page 12, line 13. 'The applicability' to what?

47) Page 12, line 17. Please replace 'fitted' with 'modeled' or a more exhaustive explanation of the way the fit was performed.

48) Page 12, line 18. Please replace 'similar' with 'similarly'.

---

## Author Comment (AC1) · 30 Sep 2020

**General comments**

The manuscript presents a statistical analysis of polarimetric radar data and in-situ (disdrometer)

observations collected over several years in Australia. The topic is relevant for the readership of AMT and the availability of such dataset is very important for future studies. I appreciate that data (and code) are made available. My concerns about this study are related to its objective, in my view not clearly defined, and to the methodological aspects that should be more thoroughly presented. In particular, I expect to see a critical discussion of a few crucial points (comparison of point-measurements with radar volumes, beam broadening effects, gridding of polar data, see the points listed below). I overall recommend a major revision of the manuscript.

We thank the reviewer for their valuable comments. Their suggestions greatly improved the readability of the paper. In particular, the introduction now makes it clearer that the retrieval development and statistical analysis are useful for providing long-term datasets for model validation. In response to other reviewers, we have also revised the PCA section to be easier to interpret by presenting a cross-covariance matrix between the original radar moment and principal component phase space.

**We also corrected many typographical and grammar mistakes. We have addressed the reviewer's specific concerns below.**

1. The stated objective of this manuscript is to provide (improved) information for validation of global circulation models. However, i do not see any tentative in this direction. The research presented here is a statistical analysis of disdrometer data and radar data in a tropical region. The goal is therefore not clearly defined and it needs some rephrasing / revision.

**We have added text in the second and fifth paragraphs of the introduction emphasizing that developing improved rainfall estimates is useful for GCM validation. The goal of the paper is emphasized in the first paragraph of page 4:**

"Creating accurate multidecadal, climate-research quality rainfall rates datasets at TWP Darwin at C- and X-band, as men-tioned previously, is useful for evaluating and improving model predictions. Lately, more radar rainfall estimators at shorter wavelengths have been developed. However, these estimators use data from relatively short field campaigns or a handful of case studies of extreme events. In this regard, these efforts are valuable but potentially not wellmatched to the challenges of creating multidecadal datasets at TWP Darwin with a mixture of typical and extreme rainfall events. This therefore stresses the importance of further assessing R retrievals for CPOL and other ARM radars at the ARM TWP site for developing such longterm datasets. To accomplish this task, this study uses four years of co-located two dimensional video disdrometer (VDIS) and CPOL data at the ARM TWP site, providing a longer and therefore hopefully more representative dataset than used in prior Darwin-based studies."

**More detailed specifics of each change are noted in the annotated version of the manuscript with the list of changes made.**

2. Should also the distance with respect to the radar be taken into account in your analysis? With in mind attenuation, beam broadening, partial beam filling, it is an important parameter. See also specific comment 3 below.

**The CPOL radar has a 1 degree beamwidth.**

**The range of the gate considered in the comparison from the CPOL radar is 30 km. We now state this in lines 32 to 35 of page 4:**

"At 30 km range, the gate dimensions are 250 m by 260 m, much smaller than a convective cell so the effects of nonuniform beam filling should be minimal. In addition, R estimation errors at C-band due to beam broadening are on the order of 0.2 mm/hr at 30 km range (Gorgucci and Baldini, 2015)."

While we acknowledge that analyses of the impact of range on rainfall retrievals are very important, such analyses require a wider rain gauge network than what was available at the ARM TWP site.

We state that we correct for potential (differential) attenuation on lines 9-12 of page 5: "The Z-PHI approach provides an estimate of the specific (differential) attenuation ( $A_{dr}$ )  $A_h$  as a linear function of  $\varphi_{dp}$  that varies depending on the presence of convective "hot spots" (Gu et al., 2011). The  $A_{dr}$  and  $A_h$  are then integrated along the ray to provide the (differential) attenuation corrected  $Z_h$  and  $Z_{dr}$ ."

3. Section 2.2: i cannot understand a few things in this section. How many disdrometers are used? Where is/are the video-disdrometer(s?) located (a map will help)? . Are they co-located with the radar? Are they distributed in a network? According to which strategy and which assumptions are volume-based radar measurements compared with point-based disdrometer measurements?

Please adapt this section in order to provide the necessary (and very important) information: it is difficult to provide a useful review and relevant suggestions when this aspect is not clear.

**We have added a figure showing a map of the location of CPOL and the video and JW disdrometers with discussion in Section 2.**

We have also add a sentence in Section 2 denoting our comparison strategy for the rest of the paper:

"The comparisons in this paper define the point as the average of the data in radial coordinates from the 0.5° PPI scan from the 4 gates closest to the VDIS. This covers a

horizontal distance of 0.5-1 km from the VDIS and is about 0.56 km above the VDIS at ground level and 30 km away from CPOL. This definition is chosen as it is consistent with the scales considered in past comparisons by Ryzhkov et al. (2005) and Giangrande et al. (2014a)."

**Specific comments**

1. P1: CSU, VDIS: undefined acronyms.

**Done as suggested.**

2. P2, L1: I would mention also the other fields where accurate rainfall estimation is crucial: nowcasting, alert issuing, climatology (etc.).

**We have added these extra uses in page 2, line 1 as suggested.**

3. P2, L21: I find this part slightly over-simplified. Beam broadening should be discussed and explained. While the radial resolution can be of 100m in polar coordinates, this will not be true in Cartesian coordinates when we are far from the radar. The conversion from polar data (radar) to Cartesian grid data seems to me a crucial point to discuss, especially given the goal of the manuscript: provide knowledge useful for comparison with global circulation models.

We use do not use the data in Cartesian coordinates, but rather define a point as the average of the 4 gates closest to the VDIS. In Section 2 (line 1-5, p. 5) we now state this:

"The comparisons in this paper define the point as the average of the data in radial coordinates from the 0.5° PPI scan from the 4 gates closest to the VDIS. This covers a horizontal distance of 0.5-1 km from the VDIS and is about 0.56 km above the VDIS at ground level and 30 km away from CPOL. This definition is chosen as it is consistent with the scales considered in past comparisons by Ryzhkov et al. (2005) and Giangrande et al. (2014a)."

4. P2, L35: The other side of the medal of ZDR and Kdp is that: (1) they use two channels, so there is twice the possibility that an hardware issue will affect them,
(2) Kdp is not a radar observable, but it needs to be estimated from dp: (Otto and Russchenberg, 2011; Wang and Chandrasekar, 2009; Grazioli et al., 2014),
(3) ZDR is affected by differential attenuation and it is affected by the incidence angle (Ryzhkov et al., 2005)

In response to point 1, we have verified that the two channels of CPOL were working correctly for the entire dataset. Since this is implied no text has been added.

In response to point 2, we are aware that Kdp has to be estimated from differential phase and there are sensitivities with errors that are difficult to characterize. However, we do expect that, physically, higher Kdp should indicate the presence of oblate drops which forms the physical basis for dual polarization estimators.

In response to point 3, we apply differential attenuation corrections for ZDR. In addition, rainfall retrievals typically use a constant elevation (like we do), so incidence angle should not impact the retrieval.

5. P4, L16-20: how are the various elevations combined to provide a proxy of precipitation near ground level?

**We use the 0.5 degree PPI scan as a proxy for the precipitation at ground level, as now described in page 5, lines 1-5:**

"The comparisons in this paper define the point as the average of the data in radial coordinates from the 0.5° PPI scan from the 4 gates closest to the VDIS. This covers a horizontal distance of 0.5-1 km from the VDIS and is about 0.56 km above the VDIS at ground level and 30 km away from CPOL. This definition is chosen as it is consistent with the scales considered in past comparisons by Ryzhkov et al. (2005) and Giangrande et al. (2014a)."

6. Section 4: i had from time to time some difficulties to understand where the polarimetric variables used in this section where coming from (i.e., simulated from VDIS or measured from the radar). I suggest to clarify this aspect through the manuscript, and maybe use a different notation for simulated variables (like ZH).

**We have denoted any simulated variables with a ",s" in the subscript for clarity.**

7. P12: data availability. Please provide also the link for the data archive.

**We have provided the link to ARM Data Discovery (https://www.archive.arm.gov/discovery/).**

8. Kdp: a few words about the estimation accuracy of this parameter should be

Kdp processing methods such as LP smooth the processed result. Therefore, the characterization of the measurement errors of Kdp are not amenable to simple error characterization due to this smoothing.

**References**

T. Otto and H. W. J. Russchenberg. Estimation of specific differential phase and differential backscatter phase from polarimetric weather radar measurements of rain. IEEE Geosci. Remote Sens. Lett., 8(5):988–992, 2011. doi: 10.1109/LGRS.2011.2145354.

Y. T. Wang and V. Chandrasekar. Algorithm for estimation of the specific differential phase. J. Atmos. Oceanic Technol., 26(12):2565–2578, 2009. doi: 10.1175/2009JTECHA1358.1.

J. Grazioli, M. Schneebeli, and A. Berne. Accuracy of phase-based algorithms for the estimation of the specific differential phase shift using simulated polarimetric weather radar data.

IEEE Geosci. Remote Sens. Lett., 11(4):763–767, 2014. doi: 10.1109/LGRS.2013.2278620. A. V. Ryzhkov, S. E. Giangrande, V. M. Melnikov, and T. J. Schuur. Calibration issues of dual polarization

radar measurements. J. Atmos. Oceanic Technol., 22(8):1138–1155, 2005. doi: 10.1175/JTECH1772.1.

---

## Author Comment (AC3) · 30 Sep 2020

**GENERAL COMMENT**

This paper presents an analysis of the applicability of dual-polarization rainfall relations for C-band radars. Although in general well written, with adequate reference to previous works and (mostly) clear illustrations, I found the approach presents some flaws, Specifically:

We thank the reviewer for their valuable comments. Their suggestions greatly improved the readability of the paper. In particular we have done the following:

- Reworked Section 4.2 to both be easier to interpret and to ensure that the dataset is properly normalized and that the correlation between rainfall rate and each principal component is being calculated.
- We present the PCA results as a cross-covariance matrix between the original radar observable space and the principal component phase space.
- In response to other reviewers we have also reworked the introduction to better emphasize that the retrievals in this study are useful for model-observation intercomparsion.
- We also corrected many typographical and grammar mistakes.

**We have addressed the reviewer's specific concerns below and provide an annotated version of the manuscript that shows all of the changes that were made.**

- Simulations (section 3.2): the measurement uncertainty is not considered in these simulations. Therefore, these results only show the parametric error. For actual applications, the measurement errors should be included in the simulations. For example, a two-parameter relation like R(Zh,Zdr) has lower parametric error than R(Zh), but may have a larger total error depending on the measurement accuracy of Zdr. Have the actual measurement errors of CPOL been considered somehow?

**We thank the reviewer for this insightful comment. We have first added a comment regarding the accuracy of the calibration of Zh and Zdr on lines 8-9, page 5:**

"The RCA technique calibrated Zh to 1 dBZ accuracy and for Zdr to 0.2 dB accuracy (Louf et al., 2019)."

**This level of measurement error particularly affects the utility of Zdr in light rain (page 9, lines 10-12):**

"Zdr from CPOL is questionable to use for times when  $R < 10 \text{ mm hr}^{-1}$  as it needs to be accurate within 0.1 dB, less than the quoted 0.2 dB accuracy, for providing reasonable estimates of R in light rain (Ryzhkov et al., 2005)."

**But in heavier rainfall (p. 10, line 2):**

"In addition, the 0.2 dB accuracy of  $Z_{dr}$  from CPOL is adequate for R estimation in heavier rainfall (Ryzhkov et al., 2005)."

**We also thank the reviewer for providing a more concise name "parametric error" for our results in Section 4.1. We have adopted this terminology throughout the paper.**

PCA analysis (section 4.2): this technique is used in this paper as an original contribution for application to dual-polarization radar rainfall estimation. The physical meaning of the results is not always clear. For example, in fig. 5 (panel b) it is not clear if all three lines are the same or something is missing. In panel a) it is a bit confusing to see the first component of Ah close to 0, after having seen an excellent correlation in fig. 3: : : Although I recognize that this can be related to a lack of familiarity with PCA analysis, I encourage to authors to provide more details about the analysis performed and better discussion of the results presented in fig. 5 and 6.

We have reworked this section to ensure that the PCA data are being properly interpreted. In particular, we made the following revisions:

- Our interpretation of the PCA was only factoring in the variability in the radar observable phase space, but not factoring in how these components also vary with R. We now add the extra step of calculating the correlation of rainfall rate with each principal component.
- We standardize our input feature space so that it has zero mean and unit variance to ensure that the differences in units between the variables do not impact the results.
- We now show importance matrices, or the absolute value of the cross-covariance matrix between the features in original and PC phase space, in an easier to understand format where higher numbers indicate greater importance of each variable to each principal component.

I encourage the authors to revise the manuscript, in particular the simulation and PCA analysis sections. Also, the three parts (simulation, PCA analysis, comparison with disdrometer) are treated quite independently and there is little comprehensive discussion in the final section. I would expect in the Conclusions a more in-depth discussion of the key findings and eventually contrasting results obtained with the different methods. As a specific example, I found the conclusion about Ah (it is said that it has little predictive capability) not enough supported by compelling arguments, nor it is considered the fact that several estimators exist for the estimation of Ah (and for Kdp) with quite different Behavior.

We rewrote the conclusion section to better integrate the results from the three sections by summarizing the consistent conclusions obtained from each of the three steps. Details of these edits are visible in the version of the manuscript with the changes shown.

SPECIFIC COMMENTS AND MINOR CORRECTIONS

- Units should be in Roman font (not Italic), e.g. mm/h.

**Done as suggested.**

- P2, L4: ": : :magnitude OF the diurnal cycle.."

**We have added the "of."**

- P3, L25: ".. were developed and using data.." change to ".. were developed using Data.."?

**Done as suggested.**

- P4, L28-29: "In addition, Zh and Zdr at C-band are prone to (differential) attenuation from heavy rainfall which may bias (underestimate) R". This sentence needs to be reformulated because underestimation of Zdr causes overestimation (not underestimation) of R. In the use of R(Zh,Zdr) estimator, the underestimation of Zh and Zdr due to attenuation tend to (at least partially) compensate because of the opposite sign of the Exponents.

**We have changed "underestimate" to now say "overestimate."**

- P4, L32: linear programming is used to estimate Kdp. More discussion on this specific estimation method may be needed, especially considering plots like in fig. 8: may the positive biased estimates R(Kdp) at low rain rates may be attributed to the specific behavior of the linear programming algorithm which always produces nonnegative Kdp Values?

**Due to factors such as smoothing that is inherent in LP based methods, it is difficult to characterize the potential errors in Kdp produced by such methods.**

- P5, L3: "Waldovel" -> "Waldvogel"

**Done as suggested.**

- P6, L7: "Darwin Colorado"?? Should it read "Darwin (Australia)"?

**We added a comma in between Darwin and Colorado.**

- P8, L6: normally "PDF" should read better than "p.d.f."

**We now call the "spread in the p.d.f." parametric error.**

- P10, L21: What is the distance between the radar and the VDIS? A map may be useful. It is mentioned that measurements may be affected by attenuation, so it is important to know the range from the radar.

**We have added a figure in the paper showing the positions of the radar and the VDIS.**

- P12, L6: "based off of limited.." -> "based on"?

**Done as suggested.**

- P12, L7: "retrieving rainfall retrievals". May read better: "retrieving rainfall estimates".

**Done as suggested.**

- P21, fig.1: panels b) and c) swapped

**We have fixed the caption to match the figure.**

- P25, fig.5: replace "S-band" with "C-band"?

Done as suggested.

---

## Author Comment (AC4) · 30 Sep 2020

General Comments
This manuscript presents a study of polarimetric relationships to estimate rainfall rates
at Darwin, Australia, based on radar retrievals from the C-band dual-polarization radar
(CPOL). The retrieved relationships are validated against a co-located two dimensional
video disdometer (VDIS) via statistical metrics. I find the manuscript relevant for publication
in AMT but I suggest the authors to perform major revisions related mainly to
the organization of speech and clarity of explanation throughout the manuscript.

**We thank the reviewer for their valuable comments. Their suggestions greatly improved
the readability of the paper. In addition to the responses below, other reviewers also
asked for more information regarding the experimental setup of the radar and
disdrometers, so a figure has been added showing this setup with relevant discussion. A
The principal component analysis section has also been rewritten at a reviewer's request
in a format that is easier to interpret. Details behind these edits are visible in the attached
manuscript with changes noted.**

1) For instance, in the Introduction, many of the sentences need a fundamental reorganization
(more specifics are presented in the Technical Comments). The overall
meaning is understandable but often words are missing or displaced and the flow of
the speech is negatively affected.

**We have responded to the specific comments regarding the grammar of the paper of the
reviewer as below.**

2) It is mentioned in the manuscript that the VDIS is co-located with the CPOL radar
but no picture of the area where the two sensors are located nor their coordinates are
provided by the authors. I suggest to clarify this and include a picture of the filed of
study in Darwin.

**We have added a figure in Section 2 showing the location of the radar and Disdrometer
on a map of the Darwin region.**

3) I have some issues with the description of a couple of Figures. In paragraph 3.1 is
described Figure 1. A mention is missing of the dashed and solid lines that separate
two types of precipitation, even though the selection criteria for convective-stratiform
precipitation are described later. Similar remarks apply to Figure 3 and its description
in paragraph 3.2. It would be good for a better reader understanding to mention which
curve (and in which panel) represents the data fit.

**For both of these sections, there is now more text showing the descriptions of these
lines in each paragraph.**

4) In Section 4.1, it is stated that the Ah-based estimators give the lowest spread for

R<10mm/hr and Kdp-based estimators give the lowest spread for R>10mm/hr. This is true only if a combination of different moments is not taken into account but this is not mentioned in the text.

**We have rephrased this sentence:**
**"***In Figures 5ab, the $A_h$-based estimators give the lowest parametric uncertainty, followed by $K_{dp}$ then $Z_h$-based estimators for time periods when R < 10 mm hr$^{-1}$ when only a single radar observable is considered.***"**

Specific Comments/Technical Corrections
1) Page 1, line 17. Please define 'VDIS'.

**We have changed "VDIS" to say "video disdrometer."**

2) Page 2, line 4-5. Please reformulate: 'is that the phase and magnitude the diurnal cycle of precipitation are not adequately resolved due to the parameterization of Convection'.

**We have rephrased this sentence to use active voice to make it more readable:**

**"***A known problem of many GCMs, including the U.S. Department of Energy's Earth Energy Exascale System Model (E3SM), is that GCMs do not adequately resolve the phase and magnitude of the diurnal cycle of precipitation (Golaz et al., 2019). This is due to the fact that GCMs parameterize convection rather than explicitly resolve it (Del Genio, 2012).***"**

3) Page 2, line 12-14. Please reformulate the sentence.

**This sentence has been simplified:**

*"In addition to cloud top height and hydrometeor type datasets, long term datasets of accurate rainfall accumulations and rates are also useful for evaluating or improving convective parameterizations in E3SM and other GCMs (Tang et al., 2019)."*

4) Page 2, line 15. 'Similar summary'?

**Done as suggested.**

5) Page 2, line 20. Please replace 'this' with 'a'.

**Done as suggested.**

6) Page 2, line 22. Please define 'R' in the Introduction.

**Done as suggested.**

7) Page 3, line 1. Please replace 'found in the limits of' with 'in'.

**Done as suggested.**

8) Page 3, line 5. Please replace with 'with DSD observations subject to comparable Limitations'.

**Done as suggested.**

9) Page 3, line 7. Please insert 'the' before 'aforementioned'.

**Done as suggested.**

10) Page 3, line 12. Please define 'DOE ARM'.

**Done as suggested..**

11) Page 3, line 19. Please define 'MC3E'.

**Done as suggested.**

12) Page 3, line 24. Please define 'RMSE'.

**Done as suggested.**

13) Page 3, line 25. Please remove 'and' before 'using'.

**Done as suggested.**

14) Page 3, line 32-33. I don't understand this sentence. Do you want to state the aim of this research? What are the challenges?

**We have revised this line (now lines 5-6, page 4):**
**"***Creating accurate multidecadal, climate-research quality rainfall rates datasets at TWP Darwin at C- and X-band, as mentioned previously, is useful for evaluating and improving model predictions.***"**

15) Page 3, line 34. 'Efforts' don't have access to data.

**We have removed the "efforts" language when this paragraph was reorganized.**

16) Page 4, line 7. Please replace 'for' with 'in'.

**Done as suggested.**

17) Page 4, line 7. No need to say 'rainfall rate R'. Either 'rainfall rate' or 'R' but make sure you defined 'R' previously.

**We defined R previously, and removed "rainfall rate."**

18) Page 4, line 7. Please use present tense: 'This study uses'.

**Done as suggested.**

19) Page 4, line 11. 'on these quantities' is superfluous.

**Done as suggested.**

20) Page 4, line 21. Please remove 'a' before '250 m' and '1'.

**Done as suggested.**
**.**
21) Page 4, line 30. Please define 'Z-PHI method' and include a reference.

**We have added the description at page 5, lines 8-10:**
*"The Z-PHI approach provides an estimate of the specific (differential) attenuation (Adr) Ah as a linear function of φdp that varies depending on the presence of convective "hot spots" (Gu et al., 2011)."*

22) Page 5, line 4. Are JW disdrometers less optimal for assessing dual-polarization radar efforts in lighter rain and/or small-drop conditions than? And, once again, 'efforts' are not a measurable variable. The authors mean 'radar moments' or 'meteorological Quantities'.

**We have rephrased this sentence:**
**"***However, J-W disdrometers are potentially less optimal for calculating dual-polarization radar  quantities in lighter rain and/or small-drop conditions than in heavier rain.***"**

23) Page 5, line 16. Please rephrase as 'After the application of these thresholds'.

**Done as suggested.**

24) Page 5, line 21. Incorrect reference to Wang et al. (2018) and Giangrande et al. (2019).

**Done as suggested.**

25) Page 6, line 3. I would not use the verb 'stratify' in a sentence where 'stratiform' is referred to the type of precipitation. Please use 'separate' or 'divide'.

**Done as suggested.**

26) Page 6, line 13-14. 'other datasets around the world' is too generic.

**Done as suggested.**

27) Page 6, line 17-18. Please finish the sentence starting with 'Prior studies find that tropical-oceanic cloud behaviors do not solely drive most of the surface rainfall here'. They drive what else then?

**We have rephrased this sentence:**

**"***Prior studies find that both tropical-oceanic cloud and continental cloud behaviors drive surface rainfall here.***"**

28) Page 6, line 26. Please define 'W'.

**We now define this as liquid water content.**

29) Page 6, line 34. The authors mean 'T15' here.

**Done as suggested.**

30) Page 7, line 16. Please add ',' after 'Therefore'.

**Done as suggested.**

31) Page 7, line 32-33. The sentence starting with 'A bootstrap' is a bit convoluted and needs rephrasing.

**We have rephrased this sentence:**
*Following Wang et al. (2018), Table 2 shows confidence intervals calculated from 1000 fits from 10,000 randomly chosen DSDs, with replacement, from the VDIS dataset.*

32) Page 7, line 33 – page 8, line 1. 'The width of the 95% confidence intervals of a, b, and c of each fit (Table 2) are less than 5% of the mean a, b, and c for each randomly generated fit'. Please mention some numbers in the text for a better understanding.

**We have added an approximate order of both the widths of the confidence intervals and the mean of each coefficients to this sentence.**

33) Page 8, line 6. Please define 'p.d.f.'.

**This sentence was rephrased to match with newly adopted terminology in the paper.**

34) Page 8, line 6,7,9. Be consistent with the use adverbs: 'firstly', 'secondly', and then 'Finally'.

**Done as suggested.**

35) Page 8, line 26. Please remove one instance of 'only'.

**Done as suggested.**

36) Page 8, line 29. Please remove 'use of'.

**Done as suggested.**

37) Page 9, line 28. Please replace 'possible' with 'possibly'.

**Done as suggested.**

38) Page 10, line 3. It is valid for X-band radar only based on the figure.

**This section was modified in accordance to suggestions of another reviewer with a correct interpretation of the PCA analysis. Therefore, this statement is no longer present.**

39) Page 10, line 23. I suggest to mention, for clarity, that the estimator is represented by the dashed line.

**Done as suggested.**

40) Page 11, line 8. Incorrect reference to 'Thompson et al. (2018)'
.
**We have corrected this citation to use the proper format.**

41) Page 11, line 10. Please insert a coma after 'previous studies' instead of after 'Here'.

**This comma has been inserted.**

42) Page 12, line 4. 'Algorithms using Ah' for what?

**This has been rephrased to say "R-Ah based estimators" instead of "Algorithms using Ah"**

43) Page 12, line 6. Please replace 'based off of' with 'based on'.

**This replacement has been done.**

44) Page 12, line 7. 'Therefore' between comas.

**These commas have been inserted.**

45) Page 12, line 8. Please use the past tense here: we 'used' instead of 'use'. Check through the manuscript that the tenses are consistent.

**The past tense is now used here.**

46) Page 12, line 13. 'The applicability' to what?

**Since the conclusion has been reworked to include more discussion of the three sections, this sentence has been changed to make this comment no longer relevant.**

47) Page 12, line 17. Please replace 'fitted' with 'modeled' or a more exhaustive explanation of the way the fit was performed.

**In the reworking of the conclusion, this sentence was removed.**

48) Page 12, line 18. Please replace 'similar' with 'similarly'

**This sentence has been removed from the manuscript in response to another reviewer.**